# Plastic Properties of Prestressed High-Strength Steel Reinforced Concrete Continuous Beams

**Jun Wang \* and Wendong Yang**

School of Civil Engineering, Northeast Forestry University, Harbin 150000, China; 19953808497@163.com
\* Correspondence: jun.w.619@163.com

**Abstract:** In this paper, the plastic properties of prestressed high-strength steel reinforced concrete (PHSSRC) beams are studied, by performing static load tests on eight built-in Q460 and Q690 pre-stressed high-strength steel reinforced concrete continuous beams and one built-in Q355 prestressed steel reinforced concrete (PSRC) continuous beam. The design parameters of the specimens were the steel strength grade, the steel ratio, the comprehensive reinforcement ratio, and the stirrup ratio. The failure mode, load–deflection curve, internal force redistribution ability, curvature ductility, plastic hinge performance, and moment modification coefficient of continuous beams under the influence of various parameters were analyzed to measure the plastic performance of the continuous beams. These results show that after the plastic hinge is formed in the middle support and mid-span of the prestressed high-strength steel reinforced concrete continuous beam, the test beam eventually becomes a rotating mechanism and is destroyed with increasing load. The built-in high-strength steel can significantly improve the bearing capacity of the specimen, and the maximum increase in the bearing capacity is 37.3%. The specimen still has a high bearing capacity after reaching the ultimate bearing capacity. With a decrease in the steel ratio, the degree of internal force redistribution is deepened, the curvature ductility is improved, and the plastic performance is enhanced. Increasing the comprehensive reinforcement ratio and the stirrup ratio can improve the plastic performance of the specimen. The calculation formula of the equivalent plastic hinge zone length is proposed. The calculation formula of the moment modification coefficient, with the relative plastic rotation angle and relative compression zone height as independent variables, is established. When the relative plastic rotation angle is not greater than $0.829 \times 10^{-5}$, the moment modification coefficient increases with increasing plastic rotation angle and is not greater than 0.37. In the range of 0.3~0.4, the moment modification coefficient decreases with increasing height of the relative compression zone.

**Keywords:** high-strength steel; prestressed steel reinforced concrete; continuous composite beams; plastic properties; moment modulation

## 1. Introduction

In recent years, the requirements of architectural design have been continuously improved. High-strength steel with strength grades of Q460 and Q690 is widely used in large-span and high-rise building structures, such as the Bird's Nest, the Sony Center, and the Yokohama Landmark Tower [1], due to its high yield strength, good plasticity, and weldability [2–4]. To meet the needs of architectural design, high-performance composite structures have gradually attracted the attention of the engineering community [5]. One of the ways to form high-performance composite structures is to use high-strength and high-performance building materials instead of traditional building materials [6,7]. The application of high-strength steel to steel reinforced concrete (SRC) structures can greatly improve the mechanical performance of SRC structures [8–10]; however, with the increase in building span or load, the application of SRC structures in construction engineering is limited, because the crack width and component deflection cannot meet the requirements of normal use [11]. The prestressed steel reinforced concrete (PSRC) structure can compensate

for this shortcoming [12]. Applying prestress in SRC structures can improve the crack closure of SRC structures [13], reduce the deflection of the structure [14], and improve the utilization efficiency of component materials [15]; therefore, PSRC structures have broader application prospects. To meet the needs of the development of building structures, further reduce the section size of components, and make full use of building space, it is necessary to apply high-strength steel to PSRC structures to improve structural performance. Because most of the structural forms of buildings use statically indeterminate structures, compared with normal beams, continuous beams are more widely used in practical engineering and are suitable for plastic design methods considering internal force redistribution, which can give full play to the performance of materials and further save materials. Studying the plastic performance of prestressed high-strength steel reinforced concrete (PHSSRC) beam is helpful for the popularization and application of high-strength steel in prestressed steel reinforced concrete composite structures.

In recent years, many scholars have studied the PSRC structure. Albrecht [16] carried out a fatigue test of prestressed steel reinforced concrete composite beams, and the results showed that applying prestress to the structure can improve the fatigue resistance of steel reinforced concrete composite beams. Since most of the actual projects are statically indeterminate structures, Xie [17] carried out research on the plastic performance of prestressed steel reinforced concrete continuous composite beams and proposed the calculation formula of the length of the equivalent plastic hinge zone and the bending moment adjustment formula. Zheng [18] studied the plastic properties of prestressed concrete continuous composite beams with encased H-shaped steel and found that the length of the equivalent plastic hinge zone of the specimen was less than 0.64 times the effective height of the beam section. When the relative plastic rotation angle was less than $0.817 \times 10^{-5}$, the moment modulation coefficient increased with increasing plastic rotation angle and was not more than 0.44, and the moment modulation amplitude of the PSRC structure was given. Scholars [19–22] have also studied the plastic properties of PSRC beams and proposed the amplitude of bending moment modulation. In addition, some scholars [23–29] analyzed the failure mode, crack development, and stiffness change of PSRC members and explored the deformation capacity of PSRC beams based on the stiffness change of the section. Based on the study of the deformation capacity of PSRC beams, scholars from various countries [30–33] studied the seismic performance of PSRC frames and analyzed the formation position and rotation ability of plastic hinges in PSRC frames. As prestressed technology improves crack resistance, it also affects the plastic deformation ability of the structure; therefore, to better apply the PSRC structure to practical engineering, many scholars [34–39] have studied the displacement ductility and plastic deformation ability of PSRC beams. To meet the increasing design requirements, based on the above research, some scholars have discussed the feasibility of replacing traditional building materials with high-strength and high-performance building materials and carried out research on prestressed steel high-performance concrete composite beams [40–42]; however, there are few studies on the plastic performance of prestressed high-strength steel reinforced concrete composite beams, and the relevant specifications are still unknown, which limits the application of prestressed steel concrete composite structures in practical engineering.

Therefore, flexural tests of prestressed high-strength steel reinforced concrete continuous beams with built-in Q460 and Q690 were carried out to analyze their plastic properties and reveal the development law of plastic internal force redistribution, to provide a reference for the popularization and application of high-strength steel in prestressed steel reinforced concrete structures.

## 2. Experimental Program

### 2.1. Test Specimens

A total of nine continuous beams were designed in this test, with a beam length of 6000 mm and a section size of 250 mm $\times$ 340 mm. The longitudinal steel bars were HRB400 steel bars, the stirrups were HPB235 steel bars with a diameter of 8 mm, and the steel was

welded H steel. An unbonded prestressed tendon with a nominal diameter of 15.2 mm was used, and the prestress was applied using the posttensioning construction process. The thickness of the concrete protection was 20 mm. The built-in steel reinforcement cage skeleton is shown in Figure 1. The high prestressed concrete beams had a good displacement ductility [43], and the PPR of the test beams was 0.71~0.52; PPR is the prestress degree of the test beams. The main parameters of the specimen were the steel strength grade, the steel ratio, the comprehensive reinforcement ratio, and the stirrup ratio. The specimen number and description are shown in Figure 2. The geometric dimensions and section reinforcement of the test beam are shown in Figure 3. The design parameters of the test beam are shown in Table 1.

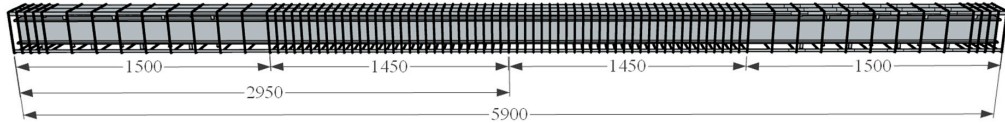

**Figure 1.** Built-in steel reinforcement cage skeleton.

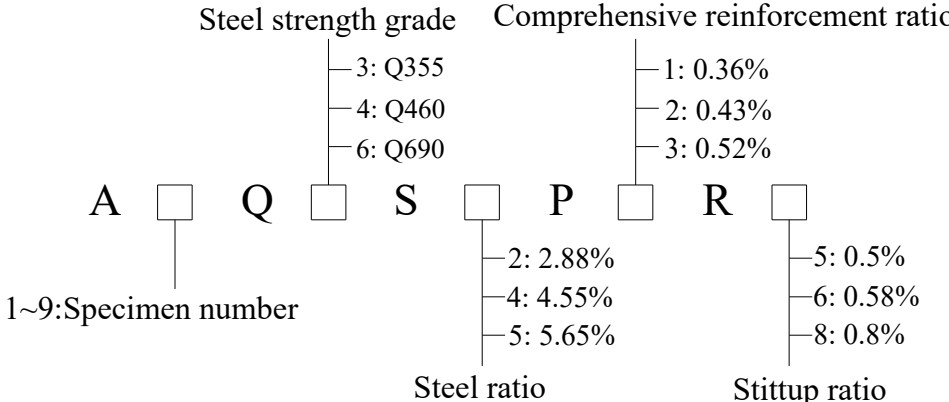

**Figure 2.** Labeling rule of specimens.

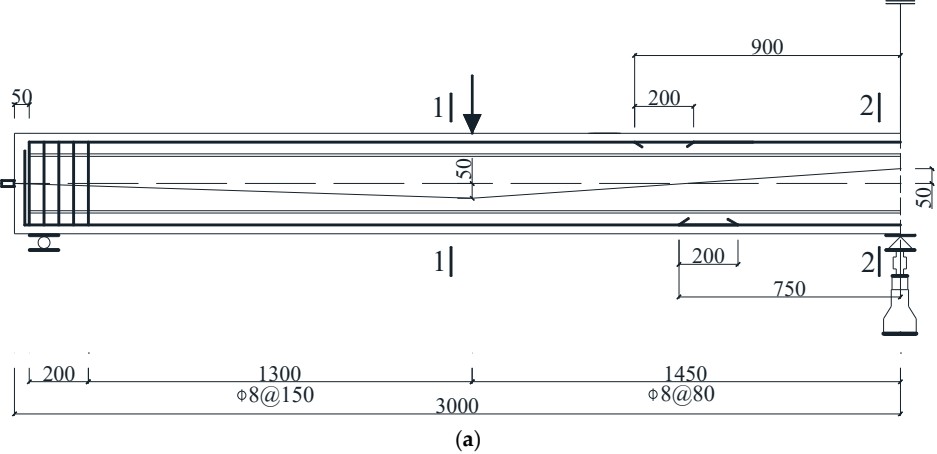

(**a**)

**Figure 3.** *Cont*.

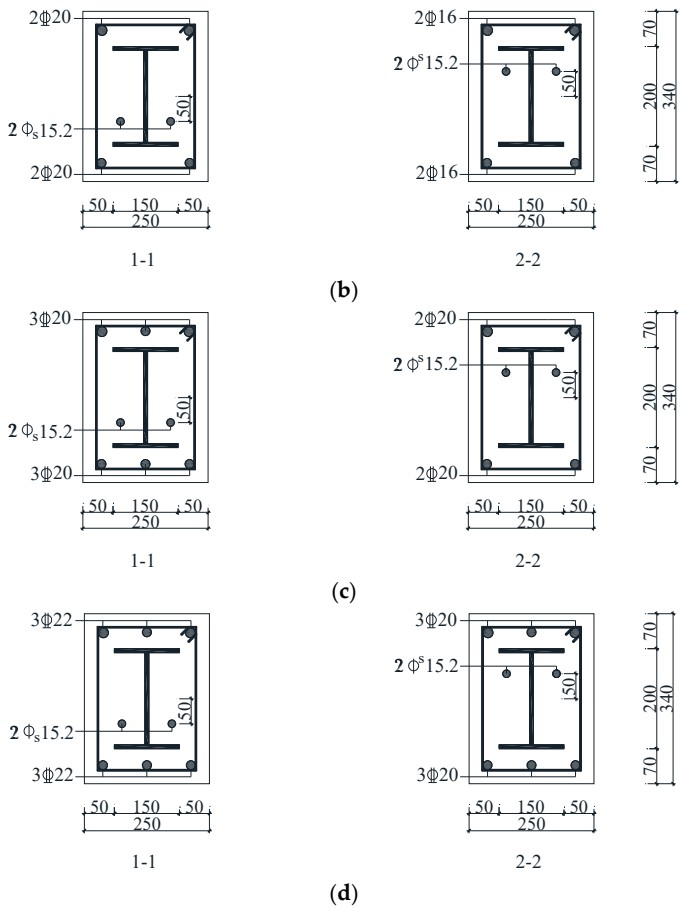

**Figure 3.** Cross-section and reinforcement of specimens (dimensions are presented in millimeters). (**a**) PSRC beam specimen dimension in this study (the figure shows half of the span of a continuous beam; the other half span is symmetrical); (**b**) Cross-sectional dimensions and reinforcement of specimen A-2; (**c**) Cross-sectional dimensions and reinforcement of specimens A-1,3,5–9; (**d**) Cross-sectional dimensions and reinforcement of specimen A-4.

**Table 1.** Parameters of the specimen.

| Specimen Number | Steel Size/mm ($b_f \times h \times t_1 \times t_2$) [1] | Steel Strength Grade | $P_s$(%) [2] | $\zeta_p$(%) [3] | $P_v$(%) [4] |
|---|---|---|---|---|---|
| $A_1Q_3S_4P_2R_5$ | $150 \times 200 \times 8 \times 8$ | Q355 | 4.55 | 0.43 | 0.5 |
| $A_2Q_4S_4P_1R_5$ | $150 \times 200 \times 8 \times 8$ | Q460 | 4.55 | 0.36 | 0.5 |
| $A_3Q_4S_4P_2R_5$ | $150 \times 200 \times 8 \times 8$ | Q460 | 4.55 | 0.43 | 0.5 |
| $A_4Q_4S_4P_3R_5$ | $150 \times 200 \times 8 \times 8$ | Q460 | 4.55 | 0.52 | 0.5 |
| $A_5Q_6S_2P_2R_5$ | $150 \times 200 \times 5 \times 5$ | Q690 | 2.88 | 0.43 | 0.5 |
| $A_6Q_6S_4P_2R_5$ | $150 \times 200 \times 8 \times 8$ | Q690 | 4.55 | 0.43 | 0.5 |
| $A_7Q_6S_5P_2R_5$ | $150 \times 200 \times 10 \times 10$ | Q690 | 5.65 | 0.43 | 0.5 |
| $A_8Q_6S_4P_2R_6$ | $150 \times 200 \times 8 \times 8$ | Q690 | 4.55 | 0.43 | 0.58 |
| $A_9Q_6S_4P_2R_8$ | $150 \times 200 \times 8 \times 8$ | Q690 | 4.55 | 0.43 | 0.8 |

[1] $t_1$ and $t_2$ represent the thickness of the steel flange and web, respectively, and bf and h represent the width of the steel flange and the height of the steel, respectively. [2] Steel ratio. [3] Comprehensive reinforcement ratio: $\zeta_p = \frac{\sigma_{pe}A_p + f_y A_s}{f_c b h_p}$, where $\sigma_{pe}$ is the effective prestress value of the prestressed tendon; $A_p$ is the cross-sectional area of the prestressed tendon; $f_y$ is the design value of the tensile strength of the tensile steel bar; $A_s$ is the cross-sectional area of the longitudinal reinforcement in the tensile zone; $f_c$ is the design value of the axial compressive strength of the concrete; $b$ is the width of the cross section; and $h_p$ is the distance between the resultant force point of the prestressing tendon and the compression edge of the section. [4] Stirrup ratio.

### 2.2. Materials

According to the requirements of the "metal material tensile test" (GB/T 228.1-2010) [44], the standard parts were sampled, and the mechanical properties were tested. The mechanical properties of the Q355, Q460, and Q690 steel plates used for welding the H-beam are shown in Table 2, and the mechanical properties of the longitudinal bars and stirrups are shown in Table 3. The specimens were poured with C50 concrete. According to the "Standard for Test Methods of Concrete Structures" (GB/T 50152-2012) [45], the mechanical properties of the concrete cube test block (150 mm × 150 mm × 150 mm) were tested, and the average value of the cube compressive strength was 55.6 MPa.

**Table 2.** Performance index of the steel plate.

| Strength Grade | Thickness /mm | Yield Strength /MPa | Ultimate Tensile Strength /MPa | Elongation /% |
|---|---|---|---|---|
| Q355 | 8 | 410.2 | 540.2 | 22.6 |
| Q460 | 8 | 516.7 | 606.6 | 21.0 |
| Q690 | 5 | 740.2 | 819.8 | 22.5 |
| Q690 | 8 | 787.7 | 851.1 | 19.4 |
| Q690 | 10 | 690.0 | 766.6 | 21.7 |

**Table 3.** Steel performance index.

| Steel Plate Number | Diameter /mm | Yield Strength /MPa | Ultimate Tensile Strength /MPa | Elongation /% |
|---|---|---|---|---|
| HPB300 | 8 | 358.3 | 537.4 | 28.6 |
| HRB400 | 16 | 406.8 | 569.7 | 27.8 |
| HRB400 | 20 | 433.2 | 566.8 | 29.5 |

### 2.3. Experimental Setup and Instrumentation

In the static loading test of the PSRC continuous beam, jacks are used to symmetrically apply loads at a single point in the middle of the two spans of the specimen, and the middle support is an adjustable hinge support. The test adopts graded loading, and each load is applied to 1/20 of the predicted value of the failure load. The step size of the applied load is reduced when it is close to the estimated value of the cracking load. At this time, the load of each stage is approximately 1/50 of the estimated value of the failure load. When the estimated value of the ultimate load reaches 90%, the step size of the applied load is reduced, and the load of each stage is approximately 1/50 of the estimated value of the failure load. After the ultimate load is reached, the loading is complete when the bearing capacity of the specimen decreases to 0.85 times the ultimate load. The dial gauges are arranged at the middle support and the side supports on both sides to observe the uneven settlement of the support and avoid errors. Displacement meters are arranged in both spans of the continuous beam to measure the deflection value of the test beam under various loads. Pressure sensors are installed under the middle support and both sides of the support to observe the change in the reaction force. The arrangement of the loading device and displacement meter is shown in Figure 4.

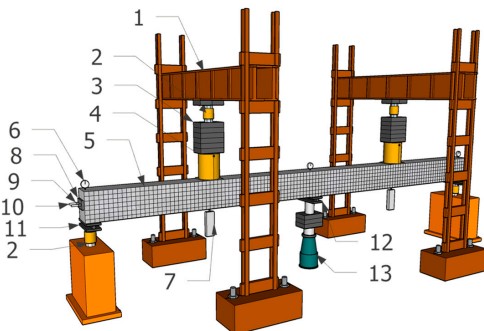

**Figure 4.** The arrangement of the loading device and displacement meter. 1 is the reaction beam; 2 is the force sensor; 3 is the iron pad; 4 is the jack; 5 is the specimen; 6 is the dial indicator; 7 is the displacement meter; 8 is the steel plate; 9 is the anchorage; 10 is the prestressed tendons; 11 is the sliding hinge support; 12 is the fixed hinge support; and 13 is the middle support jack.

### 2.4. Data Acquisition

To measure the length of the plastic hinge zone and the strain of the steel and steel bars in the plastic hinge zone, strain gauges with a spacing of 25 mm were uniformly pasted on the longitudinal reinforcement and the outer side of the upper and lower flanges of the steel, in the range of 510 mm (1.5 times the beam height) on both sides of the intermediate support of the test beam. Strain gauges were also pasted on the longitudinal reinforcement of the mid-span control section and the upper and lower flanges of the steel of the mid-span control section. The arrangement of the measuring points is shown in Figure 5.

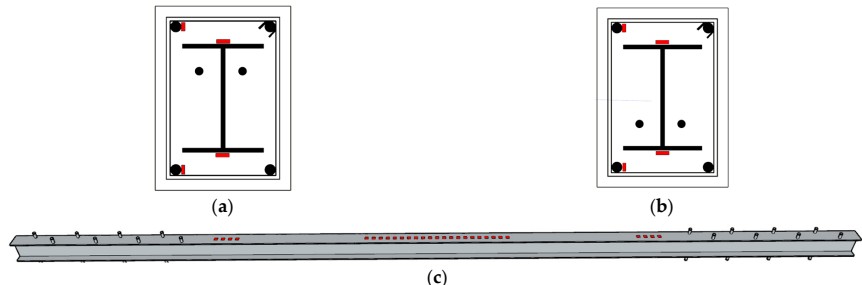

**Figure 5.** Measuring point arrangement. (**a**) Position distribution of the strain gauge in the mid-span section; (**b**) Position distribution of the strain gauge in the mid-span section; (**c**) Steel flange strain gauge.

Since the steel and steel bars are both built-in areas of the components, in order to prevent the damage of the strain gauges when pouring concrete, the strain gauges were sealed with neoprene in this test. After hardening, the strain gauges were covered and pasted with polyvinyl chloride tape. At the same time, whether the wire is effectively protected also affects the effectiveness of the strain gauge; therefore, the researchers labeled the wire and arranged it into a bundle with a nylon tie, and then wrapped the wire bundle with a PVC hose to prevent the impact damage of the concrete. The steel strain gauge used in this test was a rubber-based wire resistance strain gauge with a size of 5 mm × 3 mm. The resistance value was 120 Ω and the sensitivity coefficient was 2.05. The steel bar strain gauge used in this test was a rubber-based wire resistance strain gauge with a size of 3 mm × 1 mm. The resistance value was 120 Ω and the sensitivity coefficient was 2.12.

### 3. Specimen Failure Characteristics

In this test, the ultimate compressive strain of 3300 με was reached at the edge of the concrete in the compression zone of the control section of the intermediate support, as a sign of damage. At the beginning of the loading test, with the increase in load, the strain gauge of the tension flange of the section steel in the middle of the continuous beam first reached the yield strain of the section steel (the yield strain of the Q355 section steel was

1990 με, the yield strain of the Q460 section steel was 2475 με, and the yield strain of the Q690 type steel was 3520 με), which indicated that the plastic hinge at the section of the support in the middle of the continuous beam had begun to form. After that, as the load of the continuous beam continued to increase, the tensile flange of the steel at the mid-span section of the continuous beam reached its yield strain, which indicated that the formation of the plastic hinge at the mid-span section lagged behind. With the increase in the applied load in the mid-span of the two-span continuous beam, both the middle support and the mid-span plastic hinge rotated. The test beam eventually became a rotating system and was destroyed. The load characteristic values of the specimens under different working conditions are shown in Table 4. The typical failure mode of the specimen is shown in Figure 6a, the specimen shown in the figure is $A_6Q_6S_4P_2R_5$.

**Table 4.** The load characteristic values of the specimens under different working conditions.

| Specimen Number | $P_y$ [1]/kN | $P_m$ [2]/kN | $P_u$ [3]/kN |
|---|---|---|---|
| $A_1Q_3S_4P_2R_5$ | 432 | 695 | 735 |
| $A_2Q_4S_4P_1R_5$ | 405 | 778 | 838 |
| $A_3Q_4S_4P_2R_5$ | 455 | 805 | 875 |
| $A_4Q_4S_4P_3R_5$ | 548 | 830 | 910 |
| $A_5Q_6S_2P_2R_5$ | 480 | 795 | 855 |
| $A_6Q_6S_4P_2R_5$ | 593 | 963 | 1050 |
| $A_7Q_6S_5P_2R_5$ | 750 | 976 | 1090 |
| $A_8Q_6S_4P_2R_6$ | 608 | 966 | 1059 |
| $A_9Q_6S_4P_2R_8$ | 614 | 972 | 1062 |

[1] Yield load: The load when the steel tensile flange of the middle support section of the test beam reaches yield is used as the yield load. [2] Failure load: The load when the concrete edge of the compression zone of the middle support section is crushed is the failure load. [3] Ultimate load: Because the bearing capacity of the test beam is still improved after the failure load is reached, the maximum load that the test beam can reach during the loading process is taken as the ultimate load.

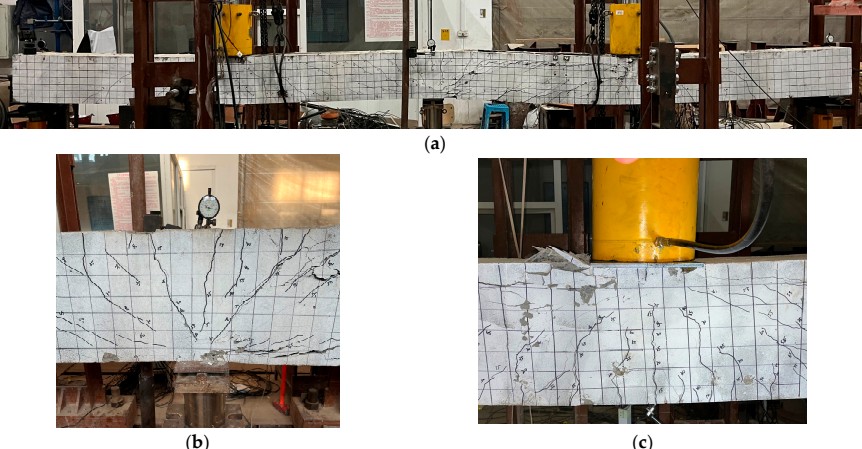

(a)

(b)                                   (c)

**Figure 6.** Specimen failure characteristics. (**a**) Specimen typical failure form; (**b**) The concrete at the control section of the middle support is crushed; (**c**) The concrete of the mid-span control section is crushed.

During the loading process, test beams A1~A9 experienced the following stages in turn: cracking of the middle support control section, cracking of the mid-span control section (0.13~0.14 $P_u$), yielding of the tension flange of the middle support control section steel, that is, plastic hinge formation (0.56~0.77 $P_u$), yielding of the tension flange of the mid-span control section steel to form a plastic hinge, crushing of the concrete of the middle support control section, continuous increasing of the load until reaching the peak load, and slow declining of the bearing capacity. Figure 6b shows that the concrete near the control section of the middle support is crushed, and Figure 6c shows that the concrete of the mid-span control section is crushed.

## 4. Test Results and Analysis

### 4.1. Load–Deflection Curve

The influence of steel strength on the load–deflection curve is shown in Figure 7. With the increase in the strength grade of the built-in steel in the continuous beam, the slope of the rising section of the load–deflection curve of the specimen increased slightly, and the linear elastic section of the rising section of the curve extended. The ultimate deflection of the built-in Q690 steel specimen was the largest, the decline rate of the load–deflection curve was the lowest, and the bearing capacity was excellent. Increasing the strength grade of the built-in steel significantly improved the ultimate bearing capacity of the specimen. The ultimate bearing capacity of the built-in Q690 steel specimen was 37.3% and 17.4% higher than that of the built-in Q355 and Q460 steel specimens, respectively. Specimen $A_5Q_6S_4P_2R_5$ had built-in Q690 steel and a steel ratio of 2.88% (Figure 7b), and specimen $A_1Q_3S_4P_2R_5$ had Q355 steel and a steel ratio of 4.55% (Figure 7a). The load–deflection curves of the two were similar, and the ultimate bearing capacity of specimen $A_5Q_6S_4P_2R_5$ was higher than that of specimen $A_1Q_3S_4P_2R_5$, indicating that built-in high-strength steel can reduce the steel consumption by 36.7% under the condition of similar bearing capacity. The strength grade of the steel in the built-in PSRC beam was improved, and the ultimate bearing capacity of the specimen was greatly improved. Under the same bearing capacity, the built-in high-strength steel can save the amount of steel.

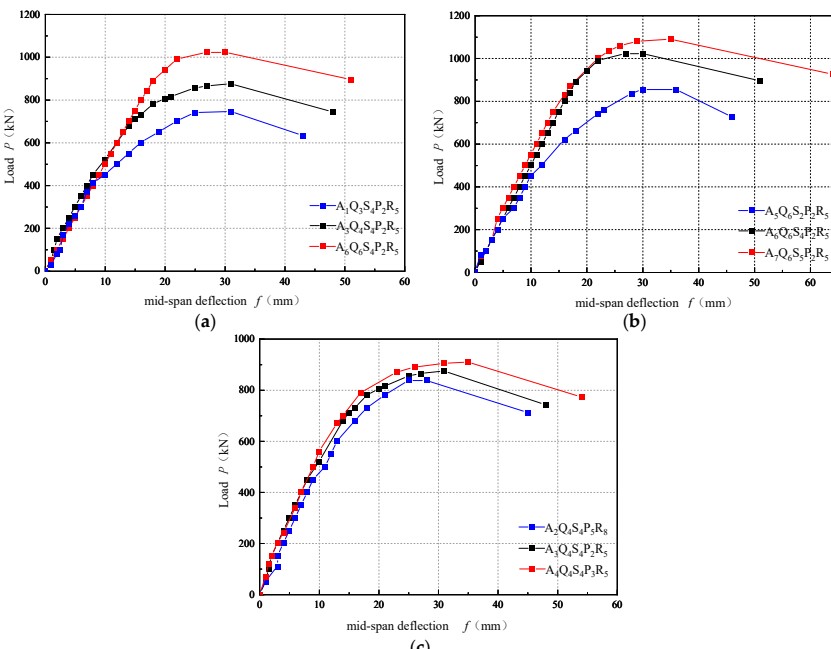

**Figure 7.** Load–deflection curve. (**a**) The influence of steel strength on the load–deflection curve; (**b**) The influence of the steel ratio on the load–deflection curve; (**c**) The influence of the comprehensive reinforcement ratio on the load–deflection curve.

With an increase in the steel ratio, the ultimate bearing capacity of the specimen increased significantly, and the ultimate deflection of the specimen increased. The influence of the steel ratio on the load–deflection curve is shown in Figure 7b. With the increase in the steel ratio, the linear elastic section of the rising section of the load–deflection curve of the specimen was prolonged, and the decline rate of the falling section of the curve was reduced. The ultimate bearing capacity was positively correlated with the steel ratio of the specimen. When the steel ratio of the Q690 steel specimen increased from 2.88% to 4.55% and 5.65%, the ultimate bearing capacity of the specimen increased by 19.6% and 27.4%, respectively.

To enhance the plastic performance of the control section of the middle bearing and the redistribution ability of the internal force of the composite beam, the comprehensive reinforcement ratio of the specimen was improved. The influence of the comprehensive reinforcement ratio on the load–deflection curve is shown in Figure 7c. With the increase in the comprehensive reinforcement ratio, the nonlinear section in the deflection curve of the specimen was prolonged, indicating that the plastic development of the section of the specimen was deepened.

### 4.2. Analysis of Plastic Internal Force Redistribution Performance

The elastic calculation value and the measured value of the side support reaction force of the test beam under the failure state are shown in Table 5. After the steel ratio of the Q690 steel specimen was reduced from 5.65% to 4.55% and 2.88%, the $R_c/R_t$ of the specimen was reduced from 0.962 to 0.935 and 0.901, indicating that the steel ratio of the high-strength steel specimen was reduced, and the internal force redistribution was sufficient. This is due to the reduction in the steel ratio, thereby reducing the tension provided by the steel during the bending process, which reduces the height of the relative compression zone of the concrete; this is conducive to the rotation of the plastic hinge and changes the internal force redistribution of the specimen.

**Table 5.** The calculated value and measured value of the side support reaction force.

| Specimen Number | $P_m$ [1]/kN | $R_c$ [2]/kN | $R_t$ [3]/kN | $R_c/R_t$ |
|---|---|---|---|---|
| $A_1Q_3S_4P_2R_5$ | 432 | 695 | 735 | 0.880 |
| $A_2Q_4S_4P_1R_5$ | 405 | 778 | 838 | 0.903 |
| $A_3Q_4S_4P_2R_5$ | 455 | 805 | 875 | 0.899 |
| $A_4Q_4S_4P_3R_5$ | 548 | 830 | 910 | 0.885 |
| $A_5Q_6S_2P_2R_5$ | 480 | 795 | 855 | 0.901 |
| $A_6Q_6S_4P_2R_5$ | 593 | 963 | 1050 | 0.935 |
| $A_7Q_6S_5P_2R_5$ | 750 | 976 | 1090 | 0.962 |
| $A_8Q_6S_4P_2R_6$ | 608 | 966 | 1059 | 0.929 |
| $A_9Q_6S_4P_2R_8$ | 614 | 972 | 1062 | 0.927 |

[1] Failure load: The load when the concrete edge of the compression zone of the middle support section is crushed is the failure load. [2] The calculated value of the side support reaction force. [3] The measured value of the side support reaction force.

In the specimens with Q460 high-strength steel, after the comprehensive reinforcement ratio changed from 0.36% to 0.43% and 0.52%, the $R_c/R_t$ of the specimens decreased from 0.903 to 0.899 and 0.885, respectively, indicating that the ability of plastic internal force redistribution was enhanced. When the height of the section relative to the compression zone was constant, the increase in the comprehensive reinforcement ratio improved the plastic development ability of the section.

### 4.3. Steel Strain Analysis and Plastic Hinge Zone Determination

The tensile strain of the steel flange can be monitored using the steel strain gauges arranged on both sides of the middle support. When the test beam reached the failure state, the strain distribution curve of the steel tensile flange in the plastic hinge section was observed. This is shown in Figure 8; the horizontal red line in all the strain distribution curves is the yield strain value of the steel tensile flange. When the compressive strain of the concrete edge of the compression zone of the control section at the middle support of the continuous beam reached 3300 με, the section curvature at this time was taken as the ultimate curvature. When the strain of the tensile flange of the test beam with built-in Q355, Q460 and Q690 steel reached 1990 με, 2475 με and 3520 με, respectively, the section curvature at this time was taken as the yield curvature.

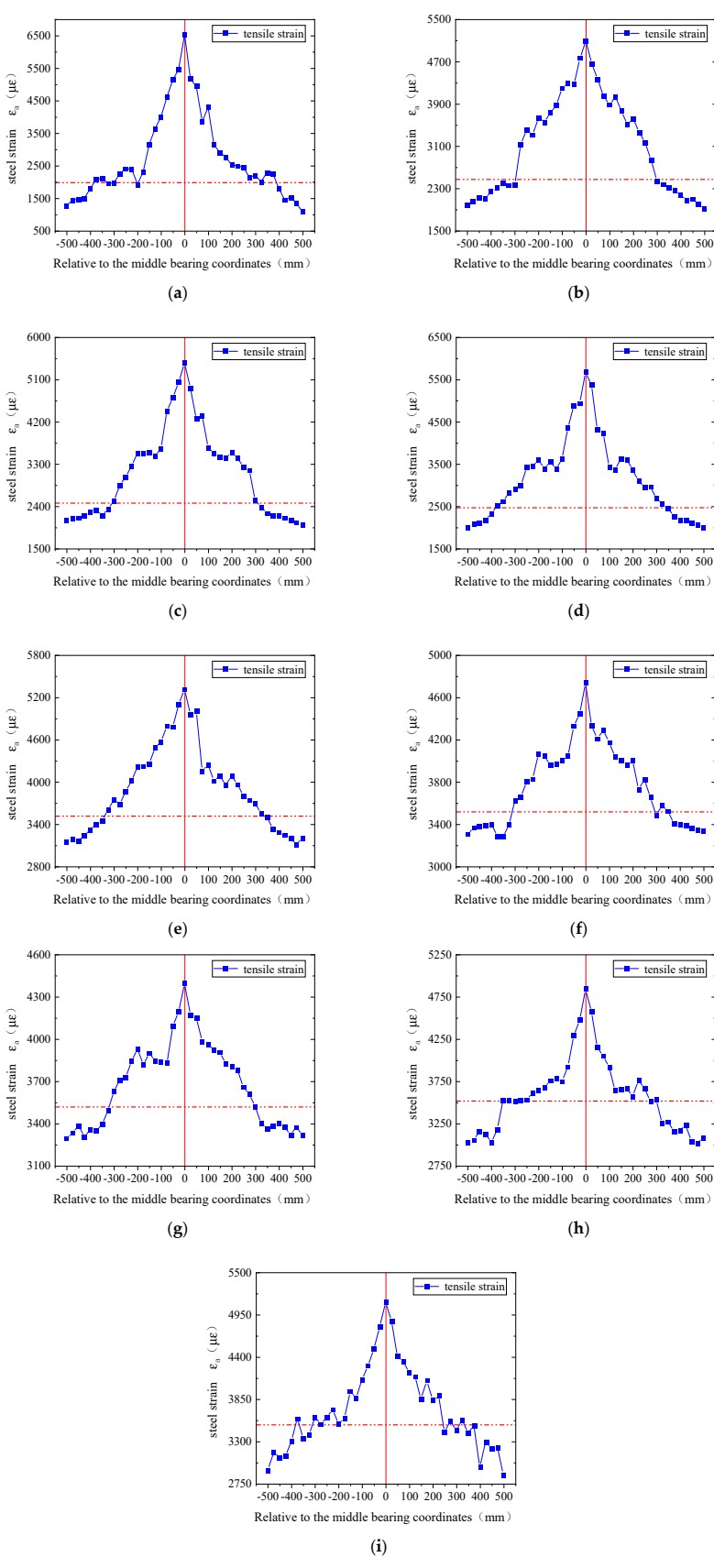

**Figure 8.** Strain distribution curve of the steel tensile flange in the plastic hinge zone. (**a**) $A_1Q_3S_4P_2R_5$; (**b**) $A_2Q_4S_4P_1R_5$; (**c**) $A_3Q_4S_4P_2R_5$; (**d**) $A_4Q_4S_4P_3R_5$; (**e**) $A_5Q_6S_2P_2R_5$; (**f**) $A_6Q_6S_4P_2R_5$; (**g**) $A_7Q_6S_5P_2R_5$; (**h**) $A_8Q_6S_4P_2R_6$; (**i**) $A_9Q_6S_4P_2R_8$.

The curvature ductility of the section is an important index of the plastic performance of the continuous beam. The curvature ductility coefficient of the section is defined as $\mu_\Phi = \Phi_u / \Phi_y$ [46], where $\Phi_u$ is the ultimate curvature of the section, and $\Phi_y$ is the curvature of the section steel when it yields. According to the plane section assumption, the ultimate curvature and yield curvature of the beam section are obtained by the strain of the upper and lower flanges of the steel and the strain of the concrete. The curvature ductility coefficient of the test beam is shown in Table 6.

**Table 6.** Curvature ductility coefficient of the test beam.

| Specimen Number | $\Phi_y$ $10^{-6}$/mm | $\Phi_u$ $10^{-6}$/mm | $\mu_\Phi$ |
|---|---|---|---|
| $A_1Q_3S_4P_2R_5$ | 16.80 | 37.86 | 2.25 |
| $A_2Q_4S_4P_1R_5$ | 19.96 | 31.80 | 1.59 |
| $A_3Q_4S_4P_2R_5$ | 20.44 | 33.14 | 1.62 |
| $A_4Q_4S_4P_3R_5$ | 21.16 | 34.52 | 1.63 |
| $A_5Q_6S_2P_2R_5$ | 23.10 | 33.64 | 1.46 |
| $A_6Q_6S_4P_2R_5$ | 24.40 | 30.62 | 1.25 |
| $A_7Q_6S_5P_2R_5$ | 25.68 | 29.60 | 1.15 |
| $A_8Q_6S_4P_2R_6$ | 24.70 | 31.32 | 1.27 |
| $A_9Q_6S_4P_2R_8$ | 25.10 | 32.47 | 1.29 |

The curvature ductility of the specimens with high-strength steel was reduced due to the large increase in the yield strain. When the steel strength grade of the composite beam is changed from Q355 to Q690, the yield strain is increased by 76.8%, and the curvature ductility of the section is decreased by 44.4%. Reducing the steel ratio of high-strength steel specimens can enhance the curvature ductility of the section, which is conducive to giving full play to the plastic deformation capacity of prestressed high-strength steel reinforced concrete continuous beams. After the steel ratio of the Q690 steel specimens decreased from 5.65% to 4.55% and 2.88%, the section curvature ductility increased by 8.69% and 26.95%, respectively. After the steel ratio of the Q690 specimen decreased from 4.55% to 2.88%, the curvature ductility of the section increased by 16.8%.

With increasing steel strength grade and steel ratio, the height of the relative compression zone of the middle bearing control section increased. The actual plastic hinge zone shortened with increasing steel strength grade and steel ratio. After the steel strength grade changed from Q355 to Q690, the actual plastic hinge zone length decreased from 385 mm to 300 mm. Increasing the comprehensive reinforcement ratio and the stirrup ratio can enhance the plastic development ability of the control section of the middle support, increase the ratio of the ultimate bending moment to the yield bending moment of the control section of the middle support, and extend the actual plastic hinge zone. After the comprehensive reinforcement ratio was changed from 0.36% to 0.52%, the actual plastic hinge zone length was expanded from 290 mm to 362 mm.

*4.4. Plastic Hinge Performance*

In this test, because the yield strength and stress area of high-strength steel were greater than those of ordinary steel bars, the curvature of the section when the tensile flange of the steel yields was taken as the yield curvature, and the curvature of the section when the compressive edge concrete of the middle support control section reaches the ultimate compressive strain ($\varepsilon_{cu}$ = 3300 $\mu\varepsilon$) was taken as the ultimate curvature.

4.4.1. The Formation and Development of Plastic Hinges

The steel strain gauges arranged on both sides of the intermediate support can monitor the formation and development of the plastic hinge of the prestressed high-strength steel reinforced concrete continuous beam. The high-strength steel at the middle support first reaches the yield strain. As the load increases, the steel flanges on both sides of the middle support gradually reach the yield strain; the farther away it is from the centre of the middle

support, the longer it takes for the steel tensile flange to reach the yield strain. On the strain distribution curve, the strain curve of the steel tensile flange in the plastic hinge section decreases stepwise from the middle support to the mid-span direction, and the strain of the steel tensile flange reaches its minimum at the end of the curve. This is because the existence of the intermediate support limits the vertical displacement of the beam so that the curvature of the control section at the intermediate support is the largest, and the strain of the steel tensile flange reaches its maximum at the centre of the intermediate support. The strain distribution curve of the tensile flange of the section steel on both sides of the middle support is stepped down from the middle support to the mid-span direction, as shown in Figure 8f.

4.4.2. Length Determination of the Equivalent Plastic Hinge Zone

When the curvature of the actual plastic hinge zone at the middle support of the continuous beam is equivalent to a rectangular distribution from the inelastic distribution, the equivalent plastic hinge zone length of the plastic hinge at the middle support of the continuous beam was obtained. Since the curvature of the continuous beam section is actually the rotation angle within the unit length of the beam, the ultimate plastic rotation angle on both sides of the middle support of the continuous beam can be calculated according to Formula (1). According to the principle of the equal ultimate plastic rotation angle, the length of the equivalent plastic hinge zone is determined and calculated according to Formula (2).

$$\theta_p = \int_0^{L_{p0}} \varphi_u(x) - \varphi_y dx \tag{1}$$

$$L_p = \frac{\int_0^{L_{p0}} \varphi_u(x) - \varphi_y dx}{(\varphi_u - \varphi_y)} \tag{2}$$

where $\theta_p$ is the limit plastic rotation angle; $L_{p0}$ is the actual length of the plastic hinge zone; $\varphi_u(x)$ represents the part of the curvature distribution curve in the plastic hinge region exceeding the yield curvature; $\varphi_y$ is the sectional yield curvature; $L_p$ is the length of the equivalent plastic hinge zone; and $\varphi_u$ is the limit curvature of the section.

After the above analysis, the section curvature of the beam section in the section on both sides of the middle support can be determined by the steel strain, and the section curvature of the section on both sides of the middle support of the continuous beam can be calculated according to Formula (3). Combined with the practical calculation method of the length of the equivalent plastic hinge zone [47–50], the equivalent rectangular method was adopted, according to the principle that the area under the inelastic curvature distribution curve in the actual plastic hinge zone is equal to the equivalent rectangular area (ensuring an equal plastic rotation angle). The equivalent plastic hinge zone length on both sides of the middle support of the test beam can be determined after the inelastic curvature is equivalent to the rectangular distribution. The measured curvature distribution and its equivalent rectangle of each test beam in the actual plastic hinge zone are shown in Figure 9, and the starting point of the ordinate in the figure is the yield curvature of the section.

$$\varphi = \frac{\varepsilon_c + \varepsilon_a}{h_a} \tag{3}$$

where $\varepsilon_a$ is the tensile strain value of the upper flange of the section steel on both sides of the intermediate support, $\varepsilon_c$ is the compressive strain value of the concrete edge of the compression zone, $\varphi$ is the section curvature of the section on both sides of the intermediate support, and $h_a$ is the distance between the centroid of the tension flange of the section steel and the edge of the concrete compression zone.

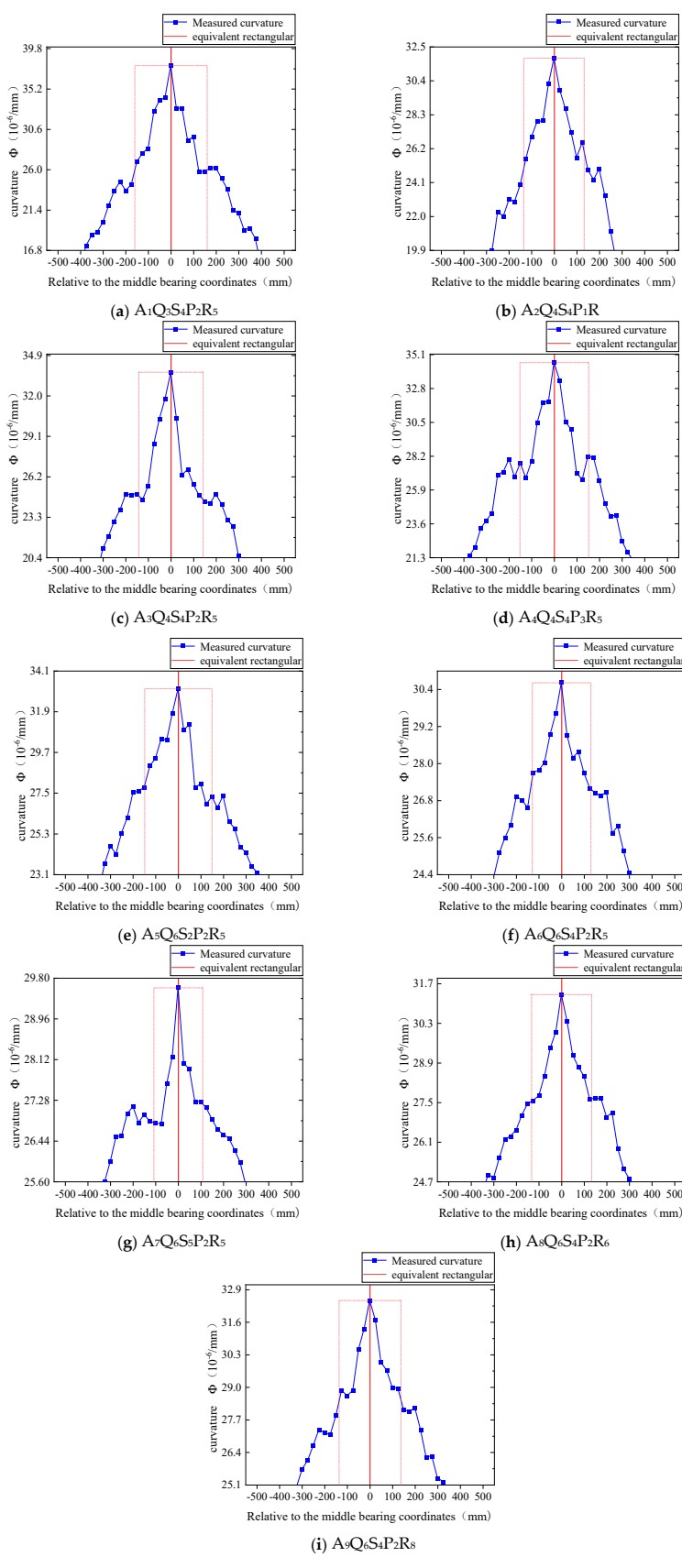

**Figure 9.** Measured curvature distribution and equivalent rectangle of the test beams. (**a**) $A_1Q_3S_4P_2R_5$; (**b**) $A_2Q_4S_4P_1R_5$; (**c**) $A_3Q_4S_4P_2R_5$; (**d**) $A_4Q_4S_4P_3R_5$; (**e**) $A_5Q_6S_2P_2R_5$; (**f**) $A_6Q_6S_4P_2R_5$; (**g**) $A_7Q_6S_5P_2R_5$; (**h**) $A_8Q_6S_4P_2R_6$; (**i**) $A_9Q_6S_4P_2R_8$.

The length of the equivalent plastic hinge zone of the PSRC beams with built-in Q460 and Q690 steel is 0.56 and 0.51 times the effective height of the beam section, respectively. After determining the length of the equivalent plastic hinge zone by drawing, the calculation formula of the ultimate plastic rotation angle can be obtained by combining Formulas (1) and (2):

$$\theta_p = (\varphi_u - \varphi_y) L_p \tag{4}$$

The measured values of the basic parameters of the plastic hinge of the middle support of each test beam, with the yield of the tensile flange of the steel as the sign of the plastic hinge, are shown in Table 7.

**Table 7.** The measured values of the basic parameters of the plastic hinge.

| Specimen Number | $\Phi_y$ $10^{-6}$/mm | $\Phi_u$ $10^{-6}$/mm | $L_p$ /mm | $\theta_p$ $10^{-3}$/rad |
|---|---|---|---|---|
| $A_1Q_3S_4P_2R_5$ | 16.80 | 37.86 | 167 | 3.517 |
| $A_2Q_4S_4P_1R_5$ | 19.96 | 31.80 | 139.8 | 1.655 |
| $A_3Q_4S_4P_2R_5$ | 20.44 | 33.14 | 142.7 | 1.812 |
| $A_4Q_4S_4P_3R_5$ | 21.16 | 34.52 | 152.4 | 2.036 |
| $A_5Q_6S_2P_2R_5$ | 23.10 | 33.64 | 149.6 | 1.577 |
| $A_6Q_6S_4P_2R_5$ | 24.40 | 30.62 | 129.6 | 0.806 |
| $A_7Q_6S_5P_2R_5$ | 25.68 | 29.60 | 118.7 | 0.465 |
| $A_8Q_6S_4P_2R_6$ | 24.70 | 31.32 | 133.6 | 0.884 |
| $A_9Q_6S_4P_2R_8$ | 25.10 | 32.47 | 137 | 1.01 |

*4.5. Calculation Formula of the Equivalent Plastic Hinge Length*

With the increase in the relative compression zone height $\xi$, the equivalent plastic hinge length $L_p$ has a decreasing trend. The corresponding relationship between the relative compression zone height $\xi$ and the equivalent plastic hinge length $L_p$ of the test beam is shown in Table 8. This is because when the relative compression zone height is large, the ratio of the ultimate bending moment to the yield bending moment is small, which reduces the length of the plastic hinge zone. According to this law, the calculation formula of the equivalent plastic hinge zone length using the relative compression zone height as the independent variable is established.

**Table 8.** The corresponding relationship between $\xi$ and $L_p$ of the test beams.

| Specimen Number | $\xi$ | $L_p$/mm |
|---|---|---|
| $A_1Q_3S_4P_2R_5$ | 0.3 | 167 |
| $A_2Q_4S_4P_1R_5$ | 0.34 | 139.8 |
| $A_3Q_4S_4P_2R_5$ | 0.338 | 142.7 |
| $A_4Q_4S_4P_3R_5$ | 0.336 | 152.4 |
| $A_5Q_6S_2P_2R_5$ | 0.329 | 149.6 |
| $A_6Q_6S_4P_2R_5$ | 0.37 | 129.6 |
| $A_7Q_6S_5P_2R_5$ | 0.4 | 118.7 |
| $A_8Q_6S_4P_2R_6$ | 0.37 | 133.6 |
| $A_9Q_6S_4P_2R_8$ | 0.37 | 137 |

The fitting curve of the equivalent plastic hinge zone length of the prestressed high-strength steel reinforced concrete continuous beam is shown in Figure 10.

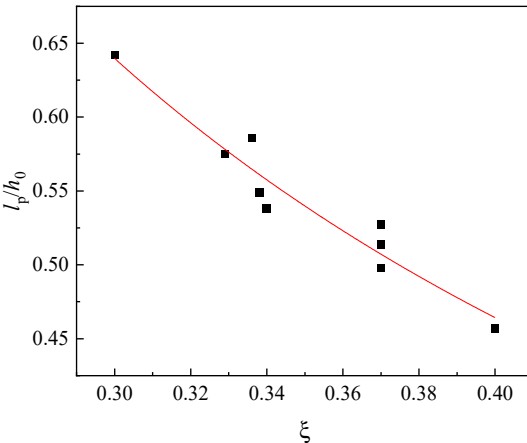

**Figure 10.** Diagram of the relationship between $\xi$ and $\frac{L_p}{h_0}$.

The fitting curve of the equivalent plastic hinge zone length of the prestressed high-strength steel reinforced concrete continuous beam can be calculated according to the following formula:

$$\frac{L_p}{h_0} = \frac{0.739}{(3.28 \cdot \xi - 0.0378)} - 0.0829 \tag{5}$$

*4.6. Analysis of the Moment Redistribution*

4.6.1. Analytical Approach

In this paper, the prestressed high-strength steel reinforced concrete continuous beam was regarded as the high-strength steel part and the prestressed reinforced concrete part [17]. The object of bending moment adjustment was the bending moment $\Delta M$, borne from the prestressed reinforced concrete part, and the bending moment adjustment coefficient was calculated according to Formula (6). The calculated elastic bending moment $\Delta M_c$ and the measured bending moment $\Delta M_t$ of the prestressed reinforced concrete part were calculated according to Formulas (7) and (8), respectively.

$$\beta = \frac{\Delta M_c - \Delta M_t}{\Delta M_c} \tag{6}$$

$$\Delta M_c = M_{load,c} - M_a \tag{7}$$

$$\Delta M_t = M_{load,t} - M_a \tag{8}$$

where $\Delta M_c$ is the calculated value of the bending moment elasticity of the prestressed reinforced concrete part; $\Delta M_t$ is the measured value of the bending moment of the prestressed reinforced concrete part; $\beta$ is the amplitude modulation coefficient of the bending moment of the prestressed reinforced concrete part; $M_a$ is the bending moment value of the section steel when the section reaches the ultimate bearing capacity state; $M_{load,c}$ is the bending moment elastic calculation value of the control section of the support under an external load; and $M_{load,t}$ is the measured bending moment value of the control section of the support under an external load.

4.6.2. Performance Analysis of Moment Amplitude Modulation

When each test beam reached the failure sign, $M_{load,c}$, $M_{load,t}$, $M_a$, $\Delta M_t$, $\Delta M_c$ and $\beta$ of the middle support control section are shown in Table 9.

**Table 9.** Moment redistribution coefficient of the test beams.

| Specimen Number | $M_{load,t}$ kN·m | $M_{load,c}$ kN·m | $M_a$ kN·m | $\Delta M_t$ kN·m | $\Delta M_c$ kN·m | $\beta$ |
|---|---|---|---|---|---|---|
| $A_1Q_3S_4P_2R_5$ | 375.19 | 290 | 146.1 | 143.9 | 229.09 | 0.37 |
| $A_2Q_4S_4P_1R_5$ | 424.13 | 348 | 161.1 | 186.9 | 263.03 | 0.29 |
| $A_3Q_4S_4P_2R_5$ | 435 | 353.8 | 160.2 | 193.6 | 274.8 | 0.3 |
| $A_4Q_4S_4P_3R_5$ | 451.31 | 353.8 | 163.9 | 189.9 | 287.41 | 0.34 |
| $A_5Q_6S_2P_2R_5$ | 429.56 | 350.9 | 161.1 | 189.8 | 268.46 | 0.29 |
| $A_6Q_6S_4P_2R_5$ | 522 | 461.1 | 248.3 | 212.8 | 273.7 | 0.22 |
| $A_7Q_6S_5P_2R_5$ | 527.44 | 493 | 279.5 | 213.5 | 247.94 | 0.14 |
| $A_8Q_6S_4P_2R_6$ | 525.26 | 458.2 | 248.3 | 207.8 | 276.96 | 0.24 |
| $A_9Q_6S_4P_2R_8$ | 527.44 | 458.2 | 248.3 | 209.9 | 279.14 | 0.25 |

The bending moment amplitude of Q460 steel specimens was higher than that of ordinary steel specimens. The bending moment amplitude of specimen A4 with a steel strength grade of Q460 was 97.51 kN·m, which was higher than that of the Q355 ordinary steel PSRC beam (85.19 kN·m).

More importantly, in the PSRC beam with a steel strength grade of Q690, reducing the steel content of the steel could improve the bending moment modulation ability of the specimen. This is because the steel content of the specimen was reduced, the height of the relative compression zone of the section was reduced, the ultimate curvature was increased, and the plastic hinge rotation ability was improved. After the steel ratio was reduced from 4.55% and 5.65% to 2.88%, the bending moment amplitude modulation coefficient increased by 31.8% and 107%, respectively.

### 4.6.3. Moment Redistribution Based on the Relative Plastic Rotation Angle

Taking the relative plastic rotation angle $\frac{\theta_p}{h_0}$ as the abscissa and the amplitude modulation coefficient $\Delta M$ of $\beta$ as the ordinate, the measured relative plastic rotation angle and bending moment amplitude modulation coefficient were placed in the coordinate system. Distribution of the $\beta$ and $\left(\frac{\theta_p}{h_0}\right) \cdot 10^5$ test points was almost linear, and the straight line obtained by first-order linear fitting is shown in Figure 11.

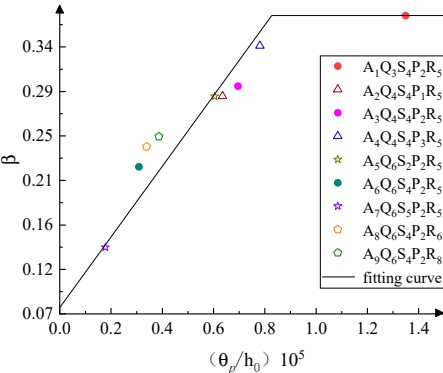

**Figure 11.** Coordinate point distribution and the fitting curve.

The bending moment amplitude modulation coefficient can be calculated using the following formula:

$$\begin{cases} \beta = 0.3558\left[\left(\frac{\theta_p}{h_0}\right) \cdot 10^5\right] + 0.075 & \left(\frac{\theta_p}{h_0}\right) \cdot 10^5 \leq 0.829 \\ \beta = 0.37 & \left(\frac{\theta_p}{h_0}\right) \cdot 10^5 > 0.829 \end{cases} \qquad (9)$$

### 4.6.4. Moment Redistribution Based on the Depth of the Compressive Zone of the Section

To predict the amplitude modulation ability of prestressed high-strength steel reinforced concrete continuous beams in practical engineering, a bending moment amplitude modulation formula with the relative compression zone height of the support as the independent variable was established. The relative compression zone height and amplitude modulation coefficient $\beta$ of the concrete in the bearing control section of the test beam are shown in Table 10.

**Table 10.** The relative compression zone heights $\xi$ and $\beta$ of the middle bearing section.

| Specimen Number | $\xi$ | $\beta$ |
|---|---|---|
| $A_1Q_3S_4P_2R_5$ | 0.3 | 0.37 |
| $A_2Q_4S_4P_1R_5$ | 0.34 | 0.29 |
| $A_3Q_4S_4P_2R_5$ | 0.338 | 0.3 |
| $A_4Q_4S_4P_3R_5$ | 0.336 | 0.34 |
| $A_5Q_6S_2P_2R_5$ | 0.329 | 0.29 |
| $A_6Q_6S_4P_2R_5$ | 0.37 | 0.22 |
| $A_7Q_6S_5P_2R_5$ | 0.4 | 0.14 |
| $A_8Q_6S_4P_2R_6$ | 0.37 | 0.24 |
| $A_9Q_6S_4P_2R_8$ | 0.37 | 0.25 |

Taking the relative compression zone height $\xi$ of the section concrete of the intermediate support as the abscissa and the amplitude modulation coefficient $\beta$ as the ordinate, the distribution of $\beta$ and $\xi$ of the prestressed high-strength steel reinforced concrete continuous composite beam in the coordinate system can be obtained. In the coordinate system, the lower envelope of the amplitude modulation coefficient $\beta$ was drawn, as shown in Figure 12.

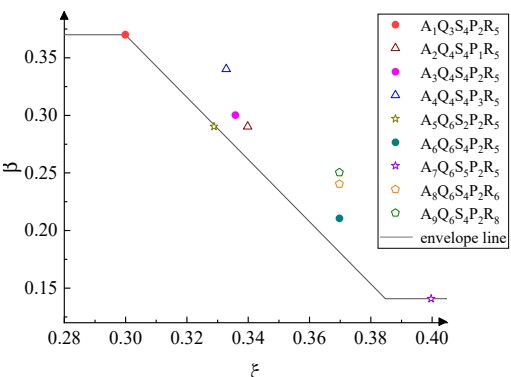

**Figure 12.** The coordinate distribution and envelope line of $\beta$ and $\xi$.

The bending moment amplitude modulation coefficient $\beta$ can be calculated as follows:

$$\beta = \begin{cases} 0.37 & \xi < 0.3 \\ 0.14 & \xi > 0.385 \\ linear\ interpolation & 0.3 < \xi < 0.385 \end{cases} \tag{10}$$

The moment amplitude modulation coefficient is calculated by Formulas (9) and (10). The moment value of the bearing section after moment amplitude modulation of the prestressed high-strength steel reinforced concrete continuous beam can be obtained using Formula (11).

$$M_d = M_a + (1 - \beta) \cdot \Delta M \tag{11}$$

## 5. Conclusions

The PSRC beams with built-in high-strength steel were tested to investigate the applicability of high-strength steel in PSRC continuous beams. These results show that the PSRC beam with built-in high-strength steel has excellent structural performance. The bearing capacity of the specimen was greatly improved under the same section size. In addition, the influence of various parameters on the plastic properties of PSRC beams with built-in high-strength steel was further analyzed. These results show that reducing the steel ratio of prestressed high-strength steel reinforced concrete continuous beams can improve the plastic internal force redistribution performance of the specimen and deepen the plastic development degree of the section. Increasing the comprehensive reinforcement ratio and stirrup ratio can improve the curvature ductility of continuous beams and enhance their plastic properties. Finally, on the basis of the above plastic performance analysis, the plastic design method of a prestressed high-strength steel reinforced concrete continuous beam was proposed, and the amplitude of the bending moment was suggested. In general, this paper analyzes the plastic properties of high-strength steel prestressed concrete continuous beams and provides a plastic design method for practical application. The main conclusions are as follows:

- Built-in high-strength steel can greatly improve the bearing capacity of PSRC beams. Compared with the bearing capacity of PSRC beams with built-in Q355 steel, the ultimate bearing capacity of PSRC beams with built-in Q460 and Q690 steel increased by 19.04% and 42.8%, respectively. In addition, by increasing the steel ratio, the bearing capacity of PSRC beams has also been significantly improved. After the steel ratio of the Q690 steel specimens increased from 2.88% to 4.55% and 5.65%, the ultimate bearing capacity of the specimens increased by 19.6% and 27.4%, respectively.
- Increasing the comprehensive reinforcement ratio and stirrup ratio in PSRC beams with built-in high-strength steel can improve the bending moment modulation ability of continuous beams. The calculation formula of the equivalent plastic hinge zone length of a PSRC continuous beam with high-strength steel was proposed.
- The moment modulation coefficient of the prestressed high-strength steel reinforced concrete continuous beam was determined, and the plastic design method of the prestressed high-strength steel reinforced concrete continuous beam was proposed. The bending moment modulation coefficient of PSRC beams with built-in high-strength steel was in the range of 0.22~0.34, and the calculation formula for bending moment modulation was established with the relative plastic rotation angle as the independent variable and the relative compression zone height of the bearing control section as the independent variable.
- The specimens with built-in Q690 steel and a steel ratio of 2.88% and the specimens with Q355 steel and a steel ratio of 4.55% reduced steel consumption by 36.7%, under the condition that the bearing capacity of the two was similar.

**Author Contributions:** J.W. proposed the topic of this study, designed the test, calculated the results according to the codes, and analyzed the data. W.Y. conducted the experiments and data aggregation, and wrote the original draft. All authors have read and agreed to the published version of the manuscript.

**Funding:** This research was financially supported by the Key Program of the Natural Science Foundation Project of Heilongjiang Province (ZD2019E001) and the Fundamental Research Funds for the Central Universities (2572019CT01).

**Institutional Review Board Statement:** Not applicable.

**Informed Consent Statement:** Not applicable.

**Data Availability Statement:** The data presented in this study are available on request from the corresponding author. The data are not publicly available due to ethical.

**Conflicts of Interest:** The authors declare no conflicts of interest.

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
