# Peer review of "Plastic Properties of Prestressed High-Strength Steel Reinforced Concrete Continuous Beams"

_applsci, doi:10.3390/app14020507_

Round 1
Reviewer 1 Report
Comments and Suggestions for Authors
Comments on the Quality of English LanguageNo such corrections were noticed, but little improvement can be done in language.
Reviewer 2 Report
Comments and Suggestions for Authors
The article presents the results of the study PSRC beams with built-in high-strength steel were tested to investigate the applicability of high-strength steel in PSRC continuous beams. In addition, the influence of various parameters on the plastic properties of PSRC beams with built-in high-strength steel was further analysed.
In my opinion this paper is interesting and complete. However, the following items are to be addressed before the manuscript can be published:
1) Figure 1: due to the size is a little unreadable, please show the half beam in a larger scale, like figure 2a.
2) Section 2.4: Please advise if the strain gauges were protected in any way from damage during concrete placement.
3) Line 176: The sentence "The failure mode of the specimen is shown in Figure 6 (a)" is misleading because it implies that the failures of all beams will be shown. In contrast, the figure shows a typical failure mode.
4) Figure 6: Please specify which specimen it represents.
5) Figure 8: Should be included as close to the reference in the text as possible; this will make it easier for the reader to understand the analysis described.
6) The main conclusions: I prefer in the form of bullet.
